# Testosterone reduces uterine contractions in vivo: Evidence for non-genomic action in rats

Saif-alnasr H. Mohammed[1,2], Mohammed Taj-Eldin Abdalla[1,3], Ayman B. Mousa[1,4], Anita Sztojkov-Ivanov[5], Kálmán Ferenc Szűcs[1]*, Róbert Gáspár[1]*

1 Department of Pharmacology and Pharmacotherapy, Albert-Szent-Györgyi Medical School, University of Szeged, Szeged, Hungary, 2 Department of Pharmacology, Faculty of Pharmacy, Omdurman Islamic University, Omdurman, Sudan, 3 Department of Pharmacology and Pharmacy Practice, Faculty of Pharmacy, Sudan University of Science and Technology, Khartoum, Sudan, 4 Department of Clinical Pharmacology, Faculty of Medicine, University of Bahri, Bahri, Sudan, 5 Department of Pharmacodynamics and Biopharmacy, Faculty of Pharmacy, University of Szeged, Szeged, Hungary

* szucs.kalman@med.u-szeged.hu (KFS); gaspar.robert@med.u-szeged.hu (RG)

## Abstract

This study aimed to investigate the non-genomic effect of testosterone (T) on uterine muscle contractility in non-pregnant and 22nd day pregnant rats, in vivo. It provides the first evidence of an in vivo T uterine relaxation effect during pregnancy. Circulating T levels were measured within 8 hours after a single intraperitoneal (ip) administration of T (10 mg/kg) by ELISA. The kinetic curves of a single dose of T (10 mg/kg of ip) were estimated for 8 hours; plasma T was measured using ELISA kits. The rapid in vivo action of T was studied by measuring contractions using strain-gauge sensors in 15-minute intervals, and the AUC was calculated. The animals received T alone (3/10/30/100/300 mg/kg ip), with flutamide (100 mg/kg ip), solvent (DMSO+Macrogol 25%+75%, 1 ml/kg ip) or normal saline (1 ml/kg) (n = 5–8/group, 160–220 g). To verify the possible mechanism of action, we also examined the uterine relaxing effect of T ($10^{-4}$ M) and nifedipine ($10^{-7}$ M) in vitro on KCl (40 mM)-stimulated contractions by cumulatively increasing the concentration of $CaCl_2$ (1–120 mM) in the organ bath. Plasma T and 4-hydroxyphenylpyruvate dioxygenase (4-HPPD) levels were measured by ELISA before and 30-minutes after administration (n = 4–8/group, 200–220 g). T had similar half-life and $t_{max}$ values in both non-pregnant and pregnant rats, however, the AUC values were higher in pregnant animals. T (30/100/300 mg/kg single ip dose) elicited a flutamide-resistant dose-dependent uterine relaxing effect in both non-pregnant and 22nd day pregnant rats. T plasma levels were proportional with the administered doses. 4-HPPD plasma levels remained unchanged in both non-pregnant and 22nd day pregnant rats after 30 min of T administration. A single dose T induces rapid, non-genomic, dose-dependent uterine relaxation via blocking of calcium effect in non-pregnant and late-pregnant rats in vivo. Based on our results, T or its analogues might be good candidates for further pre-clinical and clinical studies for uterine hyperactivity-related conditions.

**Data availability statement:** All relevant data are within the manuscript and its Supporting Information files.

**Funding:** Project No. TKP2021-EGA-32 was implemented with the support provided by the Ministry of Innovation and Technology of Hungary from the National Research, Development, and Innovation Fund, financed under the TKP2021-EGA funding scheme. The work was supported by the Stipendium Hungaricum Scholarship, University of Szeged, Hungary and by the Research Fund of Albert Szent-Györgyi Medical School, University of Szeged, Hungary. The work was also funded by the Research Fund of Albert Szent-Györgyi Medical School, University of Szeged, Hungary and University of Szeged Open Access Fund (Grant number 8261) The funders had no role in study design, data collection, analysis, decision to publish, or preparation of the manuscript.

**Competing interests:** The authors have declared that no competing interests exist.

## 1. Introduction

Conditions related to uterine smooth muscle contractions, such as premature labor, represent important concerns in clinical practice and are associated with many complications and deaths among newborns [1–6]. Androgens, estrogens, and progestins are sex steroids that play important functions in the body, primarily in the development [1–5], metabolism [1], reproduction [3–5], hemostasis [4] and modulation of various muscle tones, particularly cardiovascular [6,7], airway [8] and uterine muscles [3,9,10]. These functions are attributed to interactions with nuclear steroid receptors (SRs) that initiate a gene expression cascade; however, they also initiate fast (second to minute) non-transcriptional responses through various targets termed "non-genomic actions" [3,9,11].

Testosterone (T) is the most important hormone in the androgen family in both males and females, in males produced mainly in the Leydig cells of the testes [2,4]. Most of its actions are the result of its reduced form metabolite, 5α-dihydro-testosterone (5α-DHT), formed via 5α-reductase enzymes [2,4]. Plasma T levels in males are higher than in females [2,7,11] or in pregnant women [4,11].

T initiates both genomic and non-genomic responses; slow and long-lasting genomic actions are attributed mainly to the nuclear androgenic receptors [12,13], whereas prompt non-genomic responses result from interactions with other targets, including membrane proteins [2,9,14]. As part of its genomic actions, T can modulate the levels of androgen-dependent proteins such as 4-hydroxyphenylpyruvate dioxygenase (4-HPPD), insulin-like growth factor-binding protein 6 (IGFBP6) and fructose-bisphosphate aldolase (ALDOB), which are plasma makers of androgen activity [15].

Non-genomic actions result from interactions with several membrane proteins, including GPCRs, enzyme-linked receptors, and ion channels [6,9,16]. Protein kinase A and C, calcium ($Ca^{2+}$), MAPK, and ERK1/2 signaling pathways are stimulated nuclear independently to produce non-genomic actions [9,17,18]. GPCRC6A and ZIP9 were considered androgen targets, as both activate intracellular responses through G-proteins or MAPK signaling [4,11,12]. Cross-talk between genomic and non-genomic mechanisms is also present [9,11,18].

Previous in vitro studies demonstrated that T and its derivatives induced a concentration-dependent, rapid uterine muscle relaxant action that was resistant to androgen blocker (flutamide), protein synthesis and transcription inhibitors (cycloheximide and actinomycin D, respectively) [3,10]. However, an in vivo investigation of the non-genomic effects of androgens on uterine contractions has not yet been carried out.

Therefore, our objective was to investigate the rapid, non-genomic effect of T on uterine muscle contractility in vivo for both non-pregnant and 22nd day pregnant rats.

## 2. Materials and methods

### 2.1. Animals

The housing, handling and mating of the animals were performed as previously described [19]. In brief, healthy Sprague Dawley (SPRD) female rats were chosen for the experiment, housed in the animal facility of the Department of Pharmacology and Pharmacotherapy, Albert Szent-Györgyi Medical School, University of Szeged under

controlled temperature, humidity, and light (20–23 °C, 40–60% and 12 hrs light/dark cycle, respectively). The animals were kept on a standard Altromin 1324 rodent pellet diet (Charles-River Laboratories, Sulzfeld, Germany), with tap water available ad libitum. The animals were treated in accordance with the Directive of the European Communities Council (2010/63/EU) and the Hungarian Act for the Protection of Animals in Research (Article 32 of Act XXVIII). All experiments involving animal subjects were carried out with the approval of the Hungarian Ethical Committee for Animal Research (registration number: XIII./72/2020).

The minimum number of experimental animals required for evaluable final results was calculated using version 3.1.2 of the Power and Sample Size program with the following parameters: $\alpha = 0.0125$ (type I error); $1-\beta = 0.8$ test power; $m = 1$ (considering the number of control and treated individuals to be equal in each group); effect size $= 1.66$, which is the ratio of the difference between the means of the treatment groups and the variance of the data. The minimum number of animals per group was 4.

## 2.2. Mating and selection of rats

Mature female rats (160–220 g) in the estrus cycle were chosen based on vaginal impedance with an Estrus Cycle Monitor (IM-01, MSB-MET Ltd., Balatonfüred, Hungary). Rats whose vaginal impedance on the day of the experiment was 4.5–7.5 kΩ were chosen for non-pregnant or mating experiments. For mating, healthy SPRD male rats (240–260 g) were placed separately in a mating cage divided into 2 compartments by an automated, movable metal gate. The gate was pulled up at 4 a.m. and mating was possible within 4–5 hours. To confirm intercourse, native vaginal smears or copulation plugs were checked under the microscope at 1200× magnification. In the case of spermatozoa present in the sample or a sperm plug visible in the vagina, pregnancy was confirmed and the day of copulation was designated as the first day of pregnancy. Positive cases were housed in separate cages and used on the 22nd day of pregnancy.

## 2.3. Drugs and chemicals

T, flutamide, and nifedipine were purchased from Sigma-Aldrich (Budapest, Hungary), DMSO was from Fisher Scientific (Loughborough, UK), Macrogol 400 was purchased from MAGIlab Ltd. (Budapest, Hungary), while Rat T and 4-HPPD ELISA kits were delivered by Wuhan Fine Biotech Co., Ltd. (Wuhan, China).

## 2.4. Plasma T and 4-HPPD assessment

Plasma T levels were measured before (baseline) and after the intraperitoneal (ip) administration of a single dose of T (10 mg/kg) for non-pregnant and 22nd day pregnant rats (n = 4–8/group) using an ELISA kit with a detection range of 31.25–2000.00 pg/ml and sensitivity of 18.75 pg/ml. A 1 ml blood sample was taken from the tail vein before and after dosing (0, 5, 15, 30, 60, 120, 240, and 480 min) into tubes containing $K_2$EDTA (1 mg/tube) (BD Microtainer, Thermo Fisher Scientific Inc., Budapest, Hungary), then centrifuged (1700×g, 10 minutes, 4 °C) for the plasma. T levels were also measured before (baseline) and 30-minutes after the administration of different doses of T (3, 10, 30, 100, or 300 mg/kg) for non-pregnant and 22nd day pregnant rats. Additionally, 4-HPPD levels were measured before (baseline) and 30-minutes after the administration of the T doses (100 or 300 mg/kg) for both non-pregnant estrus rats and 22nd day pregnant (n = 6–8/group) using a commercial rat 4-HPPD Enzyme Immunoassay Kit (Wuhan Fine Biotech Co., Ltd., China), with a detection range between 31.25–2000 pg/ml and sensitivity of 18.75 pg/ml. Plasma samples were stored at −80 ºC until the assay was performed according to the manufacturer's instructions. T and 4-HPPD levels were expressed as plasma level (ng/ml), and the physiological basic T plasma values were excluded from the measured values.

## 2.5. Pharmacokinetic analysis

The pharmacokinetic parameters after T ip administration were calculated from the concentration-time profiles using non-compartmental analysis with Phoenix WinNonlin Software, version 8.5.2.4 4 (Certara Inc., Pennsylvania, USA). The maximum concentration of T in the plasma (cmax) and the time of the maximum observed concentration (tmax) were

determined from the time versus plasma concentration profiles. The elimination rate constant was estimated as the terminal slope ($\lambda_z$) by performing a linear regression analysis on the terminal phase of the logarithmic concentration versus time curve. The area from zero to infinity ($AUC_{0-inf}$) was calculated extrapolating to infinity using the equation ($AUC_{0-inf} = AUC_{0-480\,min} + c_t/\lambda_z$), where $c_t$ is the concentration measured at 480 min. The elimination half-life ($t_{1/2}$) of the terminal elimination phase was estimated using the formula $t_{1/2} = 0.693/\lambda_z$. Mean residence time ($MRT_{0-inf}$) was calculated with the formula $MRT_{0-inf} = AUMC_{0-inf}/AUC_{0-inf}$, where $AUMC_{0-inf}$ is the area under the first moment curve extrapolated to infinity. Total body clearance for extravascular administration (Cl/F) and volume of distribution based on the terminal phase ($V_z/F$) were determined using the equations $Cl/F = Dose/AUC_{0-inf}$ and $V_z/F = Dose/(\lambda_z \cdot AUC_{0-inf})$, respectively, where F is the fraction of dose absorbed.

## 2.6. Isolated organ bath study

Rats were sacrificed in a CO2 chamber with gradually increasing carbon dioxide concentrations. The uterus samples were cut from both sides of the uterine horns. After cleaning from connective and adipose tissue, 3–4-mm dissected uterine tissues were tied with silk thread and vertically mounted in an isolated organ bath filled with 10 ml of $Ca^{2+}$-free de Jongh buffer consisting of 137 millimolar (mM) NaCl, 3 mM KCl, 1 mM $MgCl_2$, 12 mM $NaHCO_3$, 4 mM $NaH2PO_4$, 6 mM glucose, the pH was adjusted to 7.40 with constant temperature (37 °C) and carbogen (95% $O_2$ + 5% $CO_2$) support. Tissues were attached to a gauge transducer (SG-02; MDE GmBH., Heidelberg, Germany), with initial resting tension of 1.5 g, the contractions were measured, recorded and analyzed with a SPEL Advanced ISOSYS Data Acquisition System (MDE GmBH., Heidelberg, Germany). The tissues were washed periodically every 15 min during the 45 min equilibrium incubation period. To achieve contraction response, KCl (40 mM) was added to each chamber for 5-min. Calcium chloride ($CaCl_2$) was added in a cumulative way (3, 10, 30, 60, 90 and 120 mM) every 3 min (Fig 1). In another set of experiments, uterine tissues were pre-treated with the T ($10^{-4}$ M) or $Ca^{2+}$ channel blocker nifedipine ($10^{-7}$ M) for 5 min before KCl (40 mM) stimulation. The contraction response percent was calculated based on KCl response, the concentration response curves were compared in the presence and absence of T or nifedipine. The samples for each experiment were collected from both sides of the uterine horns of 2 animals (8 rings/experiment) and repeated at least 3 times for each individual set of experiments.

## 2.7. In vivo contractility studies

Non-pregnant and $22^{nd}$ day pregnant rats were anesthetized using ip administration of ketamine + xylazine (36 + 4 mg/kg in 20 ml solution) in a dose of 5 ml/kg. After laparotomy, a strain gauge sensor (MSB-MET Ltd., Balatonfüred, Hungary) was fixed on the uterine surface of the rat with sutures. Rats were divided into 4 experimental groups including (1) solvent control, (2) T, (3) T + flutamide, (4) absolute control (n = 5–8 per group). T-treated animals received a single ip dose of T alone (3, 10, 30, 100 or 300 mg/kg) or with flutamide (100 mg/kg). The solvent control group received DMSO + Macrogol 400 (25% + 75%, respectively) (1 ml/kg ip); the absolute control group received physiological saline (1 ml/kg ip). The contractions were measured and recorded in 15-minute intervals before and after dose administration using the S.P.E.L. Advanced IsoSys software (MSB-MET Ltd., Balatonfüred, Hungary), and the area under the curve (AUC) was calculated. Based on the AUC changes measured for each dose, dose-response curves were plotted and the dose causing 50% maximum effect ($ED_{50}$) and maximum effect ($E_{max}$) were calculated. The results were presented as mean ± SD. The solvent or the passage of time (fatigue test) alone did not cause any significant change in contractions during the period investigated (Figs 2–4). The rats were euthanized at the end of recording with a high-dose intraperitoneal combination of anesthetics (ketamine + xylazine 360 + 40 mg/kg).

## 2.8. Statistical analysis

All data were analyzed using the Prism version 10.5 computer program (GraphPad Software Inc. San Diego, CA, USA). Values were statistically evaluated with an unpaired t-test (two-tailed), a one-way ANOVA-test (Dunnett's post hoc test), or non-linear regression. All data are expressed as means ± standard deviation (SD)

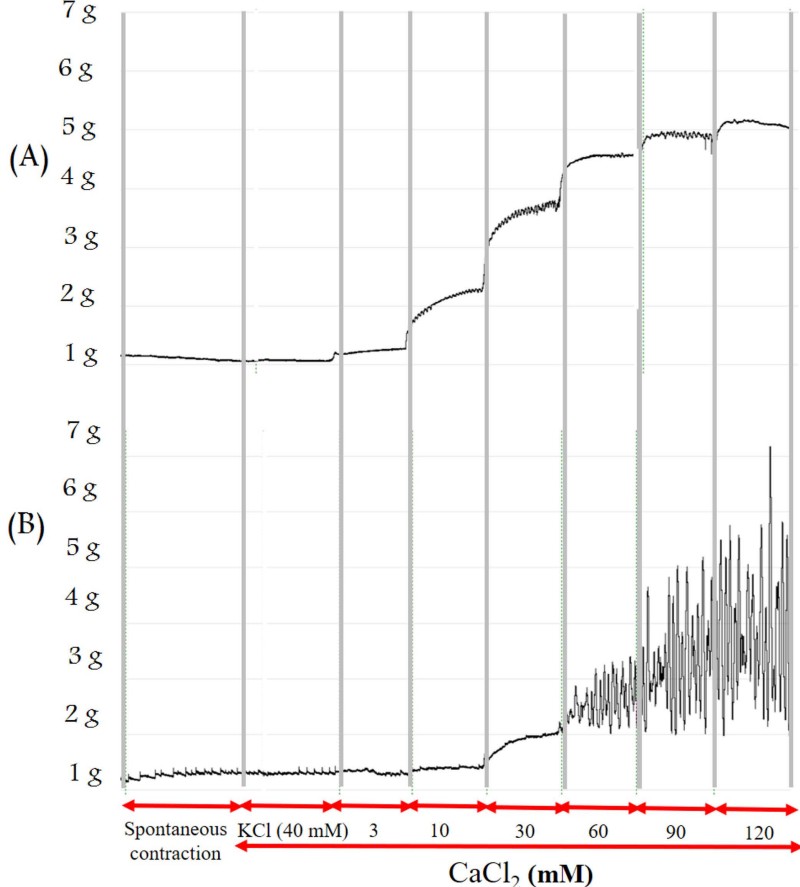

**Fig 1. The effects of CaCl2 on KCl-induced (40 mM) uterine contractions at concentrations range 3-120 mM in a cumulative manner.** Contractions were induced in the uterine rings prepared from non-pregnant and 22nd day pregnant rats. (A) a representative record for non-pregnant uterine tissue and (B) a 22nd day pregnant rat tissue.

## 3. Results

### 3.1. Testosterone pharmacokinetics in female rats

A single ip dose of T (10 mg/kg) resulted in a rapid increase in plasma T levels followed by a gradual decline in both non-pregnant and 22nd day pregnant rats (n = 5 per group). Maximum concentrations were achieved in 5 minutes in both groups, with a higher maximum value in pregnant rats (**Fig 5**, **Table 1**).

Plasma T levels increased dose-dependently in both non-pregnant and 22nd day pregnant rats 30 min after ip administration (Fig 6).

### 3.2. Results of the isolated organ bath studies

The cumulative administration of $CaCl_2$ launched and increased contractions stimulated by KCl. A concentration of T ($10^{-4}$ M) or nifedipine ($10^{-7}$ M) inhibited these contractions in non-pregnant and 22nd day pregnant rats (Fig. 7).

### 3.3. Uterine relaxation effect of T in vivo

T (30, 100 or 300 mg/kg single ip dose) induced a dose-dependent uterine relaxation effect that was not modified by the androgenic antagonist flutamide (100 mg/kg) in non-pregnant or 22nd day pregnant rats (n = 5–8 per group). The

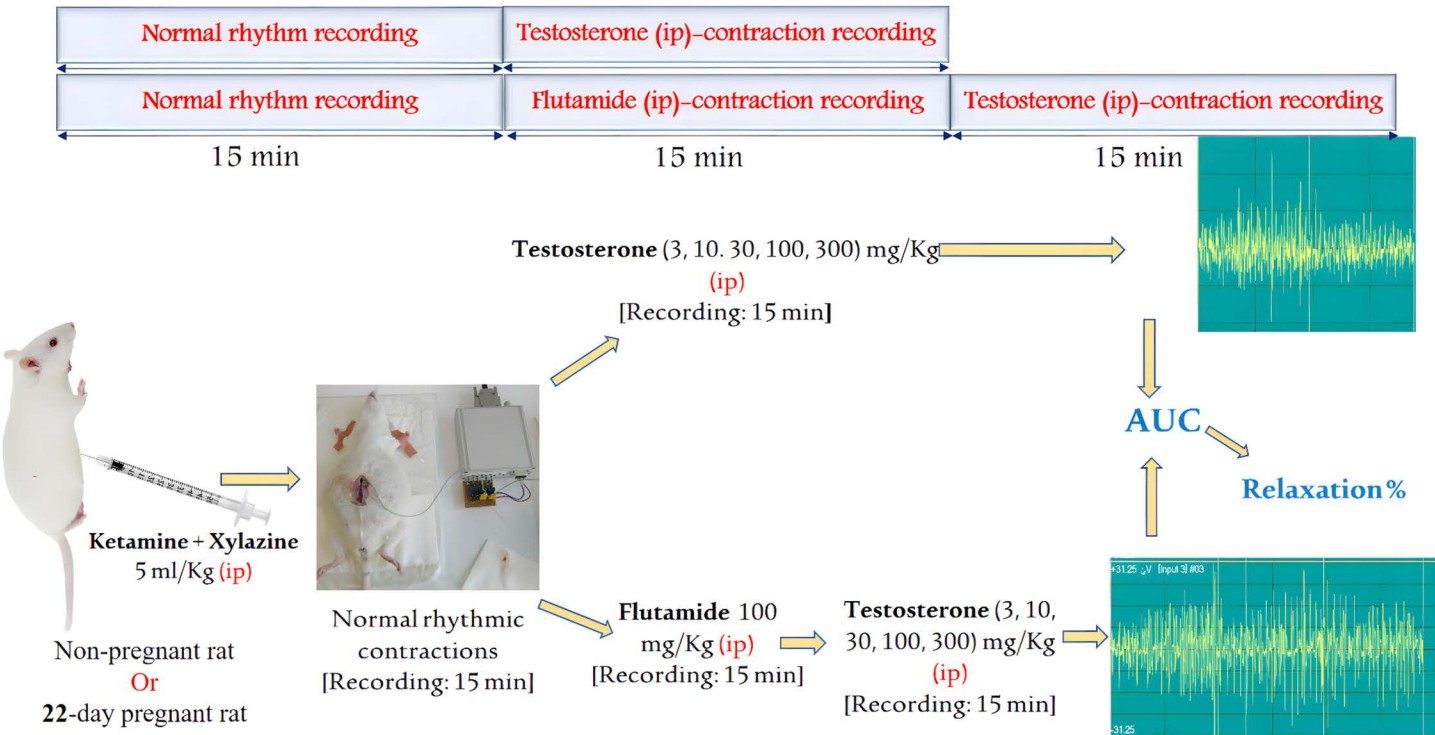

**Fig 2. The schematic diagram of the in vivo investigation of the uterine action of T in non-pregnant and 22nd day pregnant anesthetized rats.** T was administered in different doses (3, 10, 30, 100 or 300 mg/kg ip) alone and with flutamide (100 mg/kg ip) using strain gauge sensors, AUC was obtained, and the percent of relaxation was calculated and analyzed.

3 mg/kg dose was ineffective in both groups, while the 10 mg/kg dose was effective only in non-pregnant rats (**Fig 8**, **Table 2**).

### 3.4. T effect on 4-HPPD levels

Compared to baseline levels, plasma 4-HPPD levels did not show changes within 30 min after the administration of the two highest doses of T (100 and 300 mg/kg) in non-pregnant or 22nd day pregnant rats. However, 4-HPPD levels were much lower in pregnant rats (Fig 9).

## 4. Discussion

The action of sex steroids on various tissues has been examined in many studies. In addition to their classic genomic action, it was shown that they relax smooth muscles through a non-genomic (non-nuclear) pathway [15,20–24]. In vitro studies in rat uterine muscles showed that T has a muscle relaxant effect [3]. The same activity was initiated by estrogens in the vascular tissues in humans [25,26] and monkeys [27]. The non-genomic relaxant action of T was also established in human coronary arteries [28], umbilical arteries [29], peripheral vasculature and in airway smooth muscles [9,11]. However, an investigation of the non-genomic effects of androgens on uterine muscles in vivo has not yet been carried out.

Therefore, objective was to investigate the rapid, non-genomic effect of T on uterine muscle contractility in vivo in both non-pregnant and 22nd day pregnant rats. It is known that a 30 min exposition time is not enough to initiate genomic responses, therefore, all *in vivo* experiments lasted for max. 30 min. Pharmacokinetic data of T are available for male rats [30,19], but there is no information about the basic pharmacokinetic parameters of T in female rats. Therefore, such a

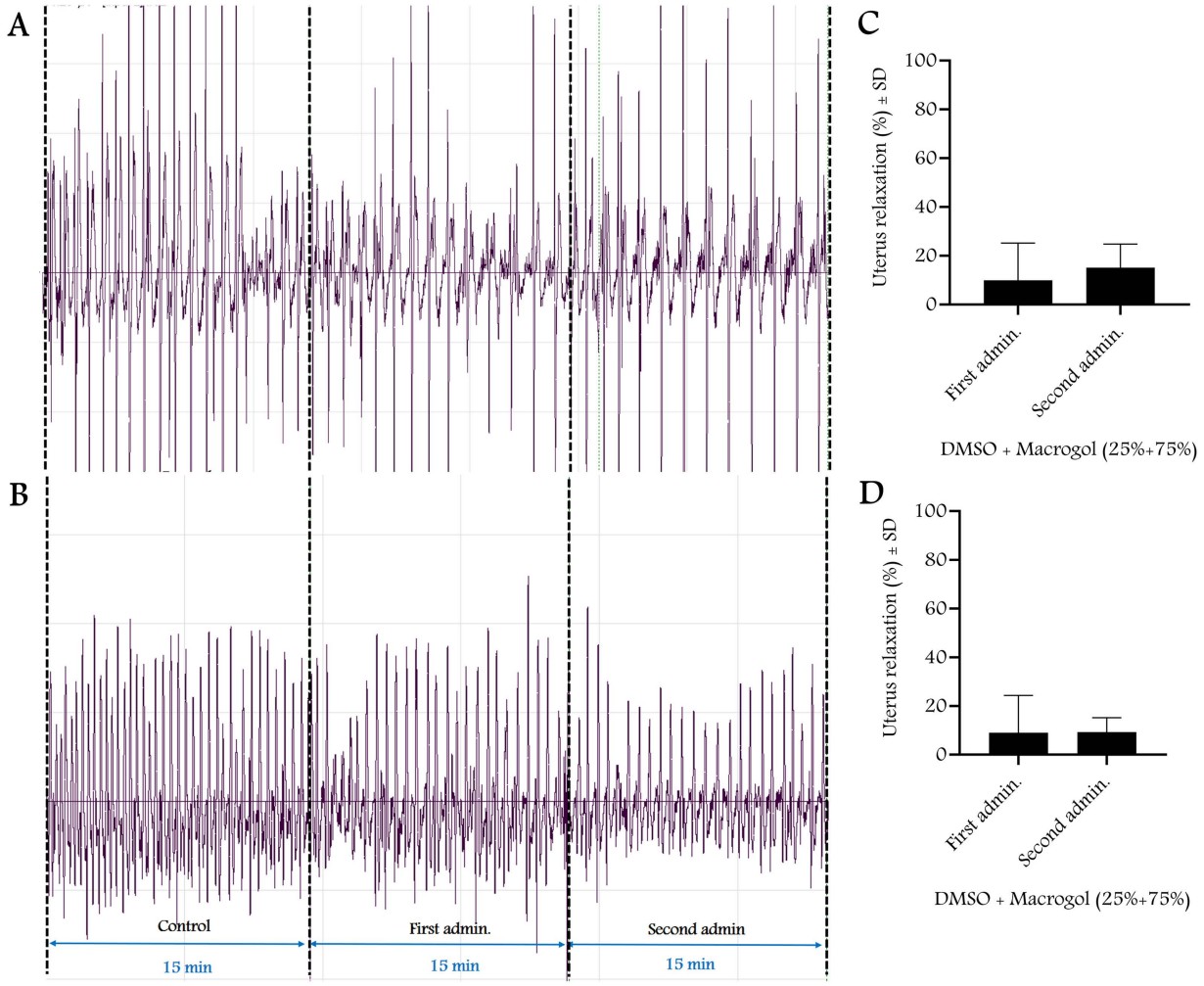

**Fig 3. The in vivo effect of solvent on the contraction of the rat uterine muscle (n=7-8 per group).** A and B are representative records of uterine contractions with strain gauge sensors, C and D represent the solvent effect in non-pregnant (A, C) and 22nd day pregnant rats **(B, D)**. Data are expressed as relaxation percent, First admin.: first administration of solvent, Second admin.: second administration of solvent dose.

study was carried out to determine whether T levels were still high enough to cause action after 30 min of administration. The T dosage range was selected based on the literature on ip systemic administration of T in male rats [20] as no data were available on T administration in female rats, especially during pregnancy. T is lipid-soluble, so it was dissolved in a solvent containing DMSO and macrogol. Due to the toxicity of this solution (e.g., hemolysis, risk of precipitation), it was not suitable for intravenous administration, so the second fastest-acting method of administration was chosen, which was ip administration.

In both non-pregnant and pregnant rats, a single 10 mg/kg injection of T ip resulted in the rapid elevation of plasma levels of T with appropriate elimination half-lives to maintain the effect for 30 min after drug administration. Interestingly, in pregnant animals the plasma time curve was higher, including the maximum value, suggesting that the absorption rate is better during pregnancy, as the half-life is similar to that of non-pregnant rats. Although basal T levels are known to be higher in pregnant rats compared to non-pregnant ones [21], this does not explain the phenomenon, as basal T levels

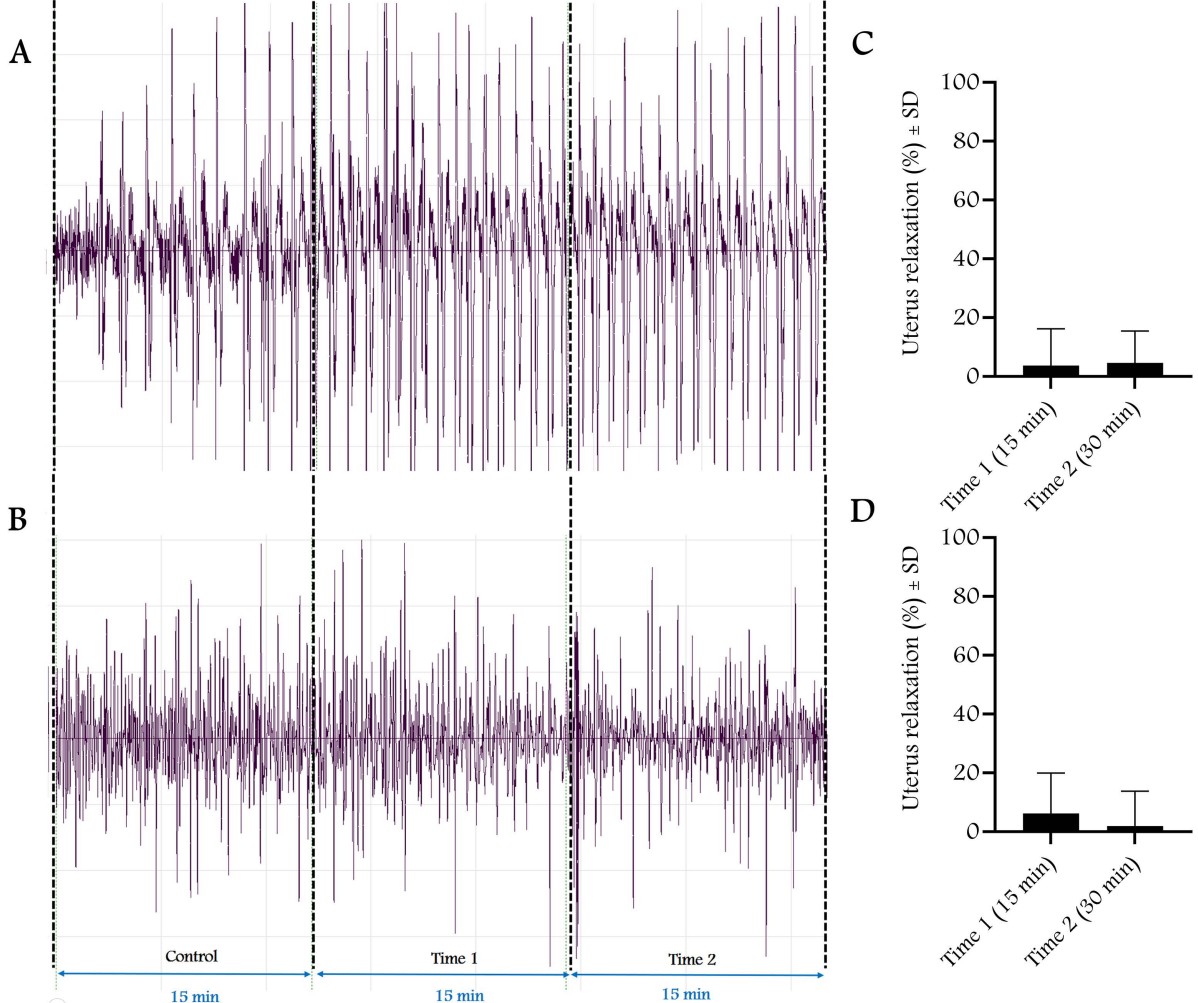

**Fig 4. The effect of time passage on the contraction of the rat uterine muscle in vivo (n = 7-8 per group).** A and B are representative records of uterine contractions with strain gauge, C and D represent the fatigue test in non-pregnant (A, C) and 22nd day pregnant rats **(B, D)**, respectively. Data are expressed as percent relaxation.

were subtracted from the measured values when calculating plasma levels. Perhaps increased abdominal circulation in pregnancy is responsible for better ip absorption [22].

Plasma T levels were also measured 30 minutes after a single ip administration of different doses to verify them at the end of the recording period during contraction studies. Plasma levels increased in a dose-dependent manner and, except for the lowest dose (3 mg/kg), significantly exceeded baseline T levels, at 10–30 folds' increase after the highest dose (300 mg/kg) demonstrating that the changes observed in contractions in both non-pregnant and 22nd day pregnant rats were related to elevated T levels.

Single doses of T induced dose-dependent uterine relaxation within 30 minutes in both non-pregnant and pregnant rats in vivo, which is consistent with previous in vitro studies on non-pregnant and pregnant uteri of rats or humans [3,10,23,24], although not all focused exclusively on non-genomic actions. Our solvent may have a relaxing effect on its own, but we have proven that it does not have a significant effect on uterine contractions.

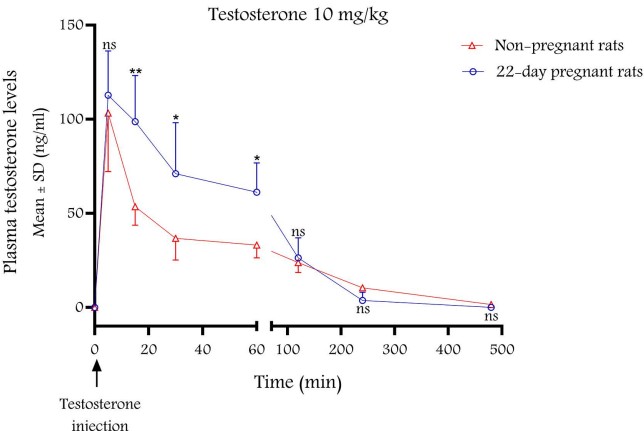

**Fig 5. Plasma T concentration-time curves after a single ip injection (10 mg/kg) in non-pregnant and 22nd day pregnant rats (n = 5 per group).** Data are expressed as mean ±SD (ng/ml). *: p < 0.05, **: p < 0.01, ns: non-significant compared to the non-pregnant value.

**Table 1. Pharmacokinetic parameters of T after single ip injection of 10 mg/kg dose in non-pregnant and 22nd day pregnant rats (n = 5 per group). Data were analysed using an unpaired t-test and are expressed as mean and SD.**

| Variable | Units | Non-pregnant | | 22nd day pregnant rats | |
|---|---|---|---|---|---|
| | | Mean | SD | Mean | SD |
| AUC | Min × ng/mL | 8464.119 | 1089.543 | 9935.925* | 2345.419 |
| Clearance (Cl) | mL/min/kg | 1196.532 | 147.012 | 1054.680 | 258.031 |
| $C_{max}$ | ng/ml | 103.225 | 30.623 | 119.521 | 19.032 |
| $t_{1/2}$ | Min | 91.026 | 10.419 | 62.353 | 23.920 |
| Elimination rate constant (Ke) | Min⁻¹ | 0.008 | 0.001 | 0.012 | 0.004 |
| $t_{max}$ | Min | 5.000 | 0.000 | 7.000 | 4.472 |

*: p < 0.05 compared to the non-pregnant value, AUC: area under the curve, SD: standard deviation

We also showed that the 30-minute time interval alone did not cause significant fatigue in the uterine response, so changes in uterine contractions were solely related to the effect of T. Nuclear androgen receptor antagonist flutamide did not modify the relaxing effect of T, providing further evidence for a non-genomic effect. 4-HPPD is a new androgen-dependent protein that is inversely proportional to the hormone. High plasma T levels induce low 4-HPPD levels [15]. Our results confirmed the reliability of 4-HPPD as an indicator of the genomic effects of T, as its levels were much lower in pregnant animals, where baseline T levels were higher than in non-pregnant rats. However, the highest doses of T (100 and 300 mg/kg) did not alter 4-HPPD levels in non-pregnant or pregnant rats, supporting the non-genomic effect of T.

We have carried out an in vitro measurement to prove the most probable non-genomic mechanism of T action via $Ca^{2+}$ channels. The T ($10^{-4}$ M) blocked the $CaCl_2$-launched contractions in a $Ca^{2+}$ free environment similarly to nifedipine ($10^{-7}$ M) strongly suggesting the role of $Ca^{2+}$ blockade in the non-genomic action of T as was suspected earlier [10,31]. However, more research is required to identify whether the uterine relaxing effect of T solely linked to $Ca^{2+}$ inhibition or steroid membrane receptors also mediate this action. It is also possible that both mechanisms are involved. The sensitivity of non-pregnant uteri was moderately higher, as 10 mg/kg T was already able to relax them, while 30 mg/kg was the lowest effective dose for 22nd day pregnant animals, although there were no differences in $ED_{50}$ values. The maximum relaxing effect of T on non-pregnant uterine muscles was moderately higher than that in

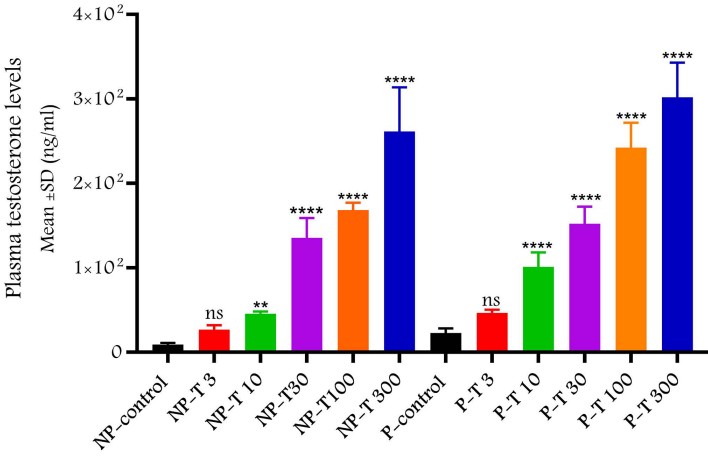

**Fig 6. Plasma T levels before and 30 min after the single dose of T administration in non-pregnant and 22nd day pregnant rats (n = 4-8 per group).** The data were analyzed using ANOVA-Dunnett's multiple comparison test compared to the control values within the group and are expressed as mean ± SD. ns: non-significant, **: p < 0.01 **** p < 0.0001, NP; non-pregnant, P; 22nd day pregnant, T; testosterone, (3, 10, 30, 100, 300 mg/kg ip).

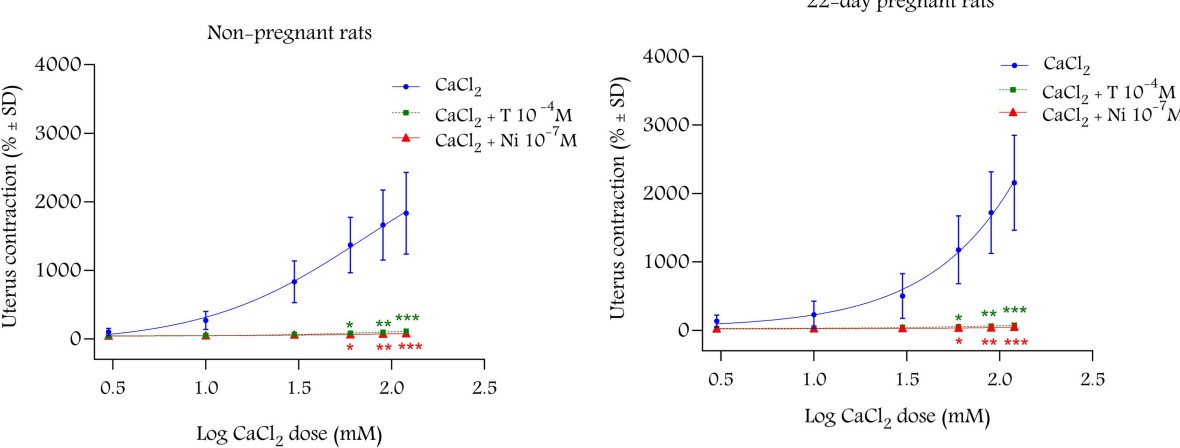

**Fig 7. The inhibitory effect of T and nifedipine on non-pregnant and 22nd day pregnant uterine tissues in vitro (n = 4-8).** The contractions were initiated with KCl (40 mM) in a $Ca^{2+}$ free buffer. Cumulative administration of CaCl2 launched uterine contractions that were completely blocked by T and nifedipine. Data were analyzed using non-linear regression and are expressed as percent of contraction mean ± SD, CaCl2: calcium chloride, T: testosterone, Ni: nifedipine.

22nd day pregnant rats. The weaker non-genomic relaxing effect of T might be explained by the higher expressions of voltage-gated $Ca^{2+}$ channels in the pregnant uterus [32]. Since one of the main mechanisms of T to relax smooth muscle is to block these $Ca^{2+}$ channels [31], higher expression of these channels may contribute to the weaker effect of T during pregnancy.

Likewise, T may have autonomous effects on uterine contractions via $Ca^{2+}$ channels and G protein-coupled membrane receptors, but these targets are in smooth muscle cells, so it seems very difficult to clearly distinguish between the autonomic and uterine smooth muscle effects of T, especially in vivo.

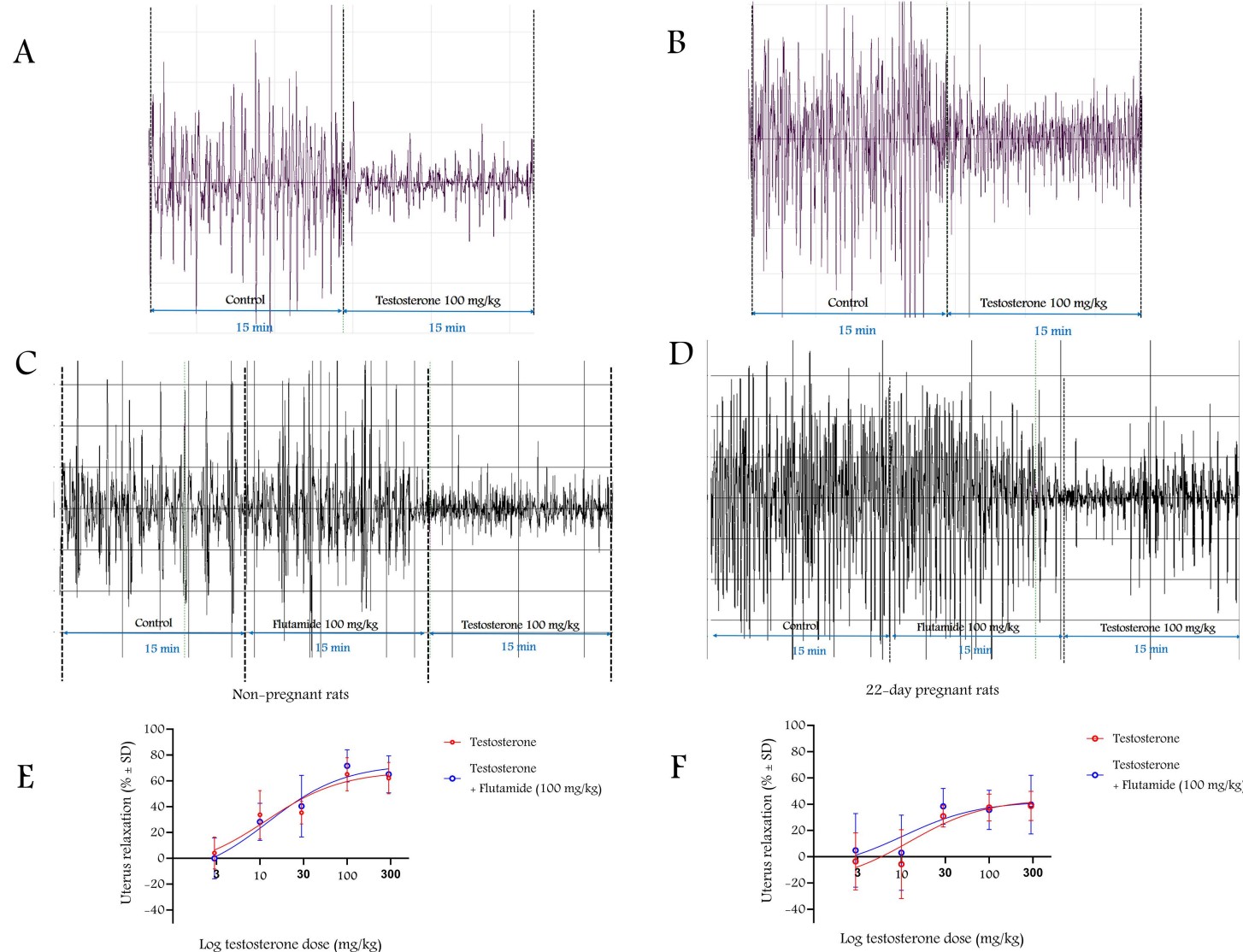

**Fig 8. The non-genomic uterine relaxing effect of T in vivo.** A, C, and B, D are representative records of strain gauge-detected uterine contractions in non-pregnant and 22nd day pregnant rats, respectively. T (ip) induced a dose-dependent but flutamide-resistant uterine relaxing effect in non-pregnant (E) and late pregnant (F) rats. Each point is the result of a single dose of T or T+flutamide, n=5-8 per group; the data are expressed as relaxation % mean±SD.

**Table 2. $E_{max}$ and $ED_{50}$ values of the uterine relaxing dose-response curves of T in non-pregnant and 22nd day pregnant rats. (n=5-8 per group).**

| Parameter | Non-pregnant rats | | 22nd day pregnant rats | |
|---|---|---|---|---|
| | T | T+Flut | T | T+Flut |
| $E_{max}$ (%) | 67.5±5.9 | 72.9±7.2 | 44.4±5.9** | 42.8±6.9** |
| $ED_{50}$ (mg/kg) | 11.7±1.9 | 12.45±1.8 | 14.3±2.3 | 11.5±2.5 |

T: testosterone, Flut: Flutamide, **: $p < 0.01$ compared to the corresponding non-pregnant value, data are expressed as mean±SD.

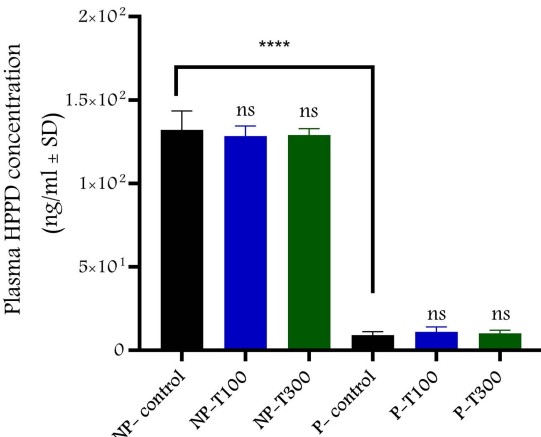

**Fig 9. Plasma levels of 4-HPPD after T administration (n = 6-8 per group).** Plasma levels of 4-HPPD were measured before and 30-minutes after the administration of T doses (100 or 300 mg/kg ip) for non-pregnant and 22nd day pregnant rats. ****; p < 0.0001, ns; non-significant, NP; non-pregnant, P; 22nd day pregnant, T100: testosterone 100 mg/kg; T300: testosterone 300 mg/kg. Data were analyzed using ANOVA-Dunnett's multiple comparison test and are expressed as mean ± SD.

The limitation of our study is that we did not investigate the long-term action of T on uterine contractions and androgen-dependent proteins. Based on the half-life of T, it may cause action for 3–4 hours, which can have a negative impact on both mothers and fetuses. More studies should focus on T derivatives that do not have genomic effects, such as 5β-dihydro-testosterone [33].

## 5. Conclusions

Single doses of T induce rapid, non-genomic, dose-dependent uterine relaxation through $Ca^{2+}$ blocking effect in non-pregnant and late-pregnant rats in vivo (Fig 10). Although the applied ip administration of T does not have direct translational significance, our study provides the first evidence of the uterine-relaxing effect of T in vivo during pregnancy. Therefore, these results may justify the development of new and more water-soluble T analogues that allow for dosing methods that are already applicable in clinical practice. Based on our results, T or its analogues may be good candidates for further preclinical and clinical studies for the potential treatment of conditions associated with uterine hyperactivity.

## Supporting information

**S1 Data. Fig 3- The in vivo effect of solvent on the contraction of rat uterine muscle.**
(XLSX)

**S2 Data. Fig 4- The effect of time passage on the contraction of the rat uterine muscle *in vivo.***
(XLSX)

**S3 Data. Fig 5 and Table 1- Plasma T concentration-time curves after a single ip injection (10 mg/kg) in non-pregnant and 22nd day pregnant rats.**
(XLSX)

**S4 Data. Fig 6- Plasma T levels before and 30 min after the single dose of T administration in non-pregnant and 22nd day pregnant rats.**
(XLSX)

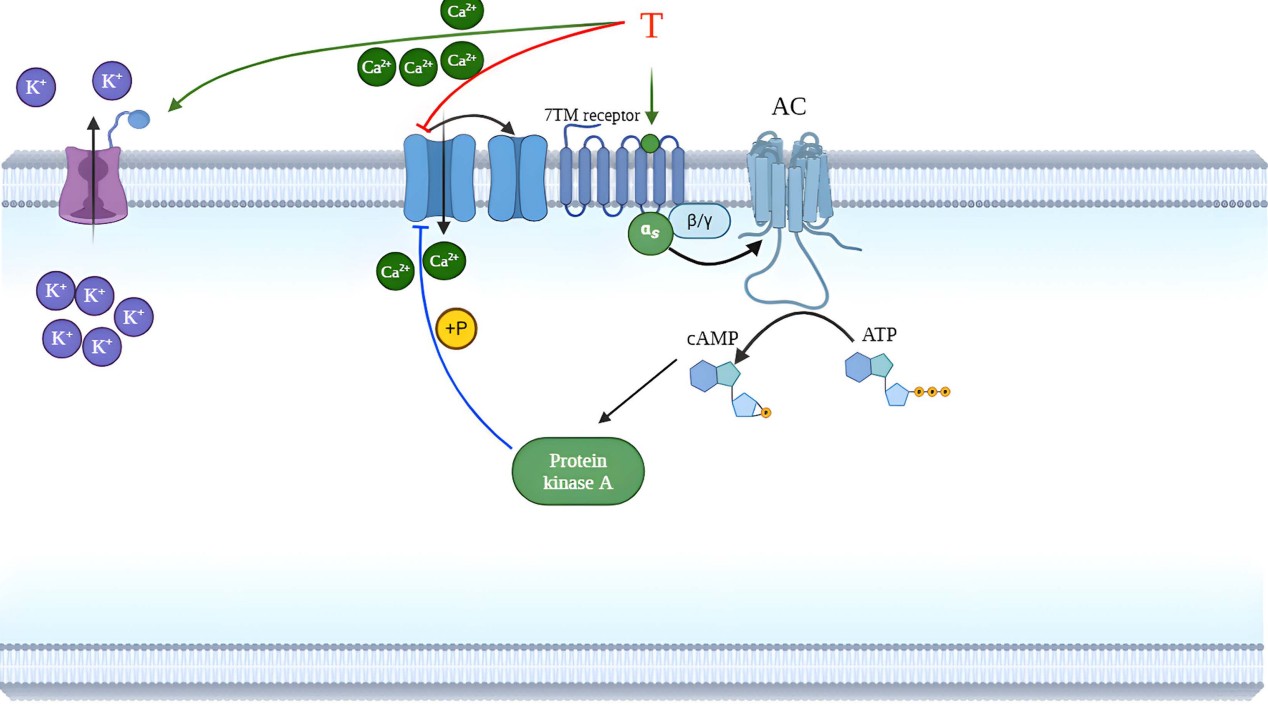

**Fig 10. Schematic diagram summarizing the effect of T on inhibiting uterine contraction.** The red line represents the inhibition demonstrated in our experiment, while the blue (inhibition) and green (activation) lines represent the mechanisms described in previous studies. T inhibits voltage dependent $Ca^{2+}$-channels having a crucial role in its uterine relaxing effect (red line). Furtherly, T stimulates the 7TM (G-protein coupled) receptors (short green arrow) and increases cAMP level of the uterine tissues which inhibits the $Ca^{2+}$ channels activity (blue line) [34]. Additionally, T activates different $K^+$ channel types in smooth muscle (long green arrow), that also leads to relaxation by reducing intracellular $K^+$ level [35]. AC: Adenylyl cyclase, ATP: adenosine triphosphate, cAMP: cyclic adenosine monophosphate, T: testosterone, 7TM: seven-transmembrane receptor (G-protein coupled receptor)".

**S5 Data. Fig 7- The inhibitory effect of T and nifedipine on non-pregnant and 22nd day pregnant uterine tissues *in vitro*.**
(XLSX)

**S6 Data. Fig 8 and Table 2- The non-genomic uterine relaxing effect of T *in vivo*.**
(XLSX)

**S7 Data. Fig 9- Plasma levels of 4-HPPD after T administration.**
(XLSX)

## Acknowledgments

The authors thank Csiszar Zoltanne for her technical assistance in the experiments.

## Author contributions

**Conceptualization:** Robert Gaspar.

**Formal analysis:** Saif-alnasr H. Mohammed, Anita Sztojkov-Ivanov, Kálmán Ferenc Szűcs.

**Funding acquisition:** Robert Gaspar.

**Investigation:** Saif-alnasr H. Mohammed, Mohammed Taj-Eldin Abdalla, Ayman B. Mousa.

**Methodology:** Saif-alnasr H. Mohammed, Mohammed Taj-Eldin Abdalla, Ayman B. Mousa, Robert Gaspar.

**Resources:** Kálmán Ferenc Szűcs.

**Writing – original draft:** Saif-alnasr H. Mohammed, Kálmán Ferenc Szűcs, Robert Gaspar.

**Writing – review & editing:** Robert Gaspar.

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
