## [Decision Letter · Decision Letter 0]

2 Jan 2026

PONE-D-25-63242Testosterone reduces uterine contractions in vivo: evidence for non-genomic action in ratsPLOS One

Dear Dr. Gaspar,

Thank you for submitting your manuscript to PLOS ONE. After careful consideration, we feel that it has merit but does not fully meet PLOS ONE’s publication criteria as it currently stands. Therefore, we invite you to submit a revised version of the manuscript that addresses the points raised during the review process.

If applicable, we recommend that you deposit your laboratory protocols in protocols.io to enhance the reproducibility of your results. Protocols.io assigns your protocol its own identifier (DOI) so that it can be cited independently in the future. For instructions see: https://journals.plos.org/plosone/s/submission-guidelines#loc-laboratory-protocols. Additionally, PLOS ONE offers an option for publishing peer-reviewed Lab Protocol articles, which describe protocols hosted on protocols.io. Read more information on sharing protocols at . Additionally, PLOS ONE offers an option for publishing peer-reviewed Lab Protocol articles, which describe protocols hosted on protocols.io. Read more information on sharing protocols at https://plos.org/protocols?utm_medium=editorial-email&utm_source=authorletters&utm_campaign=protocols..

We look forward to receiving your revised manuscript.

Kind regards,

Sayyed Mohammad Hadi Alavi

Academic Editor

PLOS One

Journal Requirements:

2. To comply with PLOS One submissions requirements, in your Methods section, please provide additional information regarding the experiments involving animals and ensure you have included details on (1) methods of sacrifice, (2) methods of anesthesia and/or analgesia, and (3) efforts to alleviate suffering.

“Project No. TKP2021-EGA-32 was implemented with the support provided by the Ministry of Innovation and Technology of Hungary from the National Research, Development, and Innovation Fund, financed under the TKP2021-EGA funding scheme. The work was supported by the Stipendium Hungaricum Scholarship, University of Szeged, Hungary and by the Research Fund of Albert Szent-Györgyi Medical School, University of Szeged, Hungary. The work was also funded by the Research Fund of Albert Szent-Györgyi Medical School, University of Szeged, Hungary and University of Szeged Open Access Fund (Grant number 8261)”

**Additional Editor Comments:**

Dear Prof Gaspar

Thank you very much for your submission to PLOS ONE. Two expert reviewers have commented on your MS. There are some concerns regarding the study design, methods and presentation of results. I agree with the reviewers that this study does not provide mechanistic information to non-genomic action of T on uterine contraction. Therefore, the MS needs a major reconstruction. In addition to the reviewers’ comments, Please find those of mine below that are required to be considered for revision. In advance, I am grateful to you for your careful consideration to all comments to revise your MS for submission.

Very best regards

Hadi Alavi, AE, PLOS ONE

Comments

1- I am so sorry, but I could not understand the meaning of Dose 1 and Dose 2 in Figure 2.

2- Please provide in-brief description of experiment in the legend of figures and title of Tables as well as sample size and Data (Mean +/- SD).

3- Please show statistical differences for T levels between pregnant and non-pregnant rats in Figure 4. It could be useful to indicate the time of T injection by an arrow.

4- I would suggest to include athe baseline HPPD level for comparisons shown in Figure 7?

5- Sensitivity and specificity of T and HPPD ELISA kits are missing; hence the detection ranges were provided.

6- Please provide the method to avoid the blood coagulation to obtain the plasma

7- Please provide a single paragraph abstract

8- Please add significance of the study at the beginning of abstract

Reviewers' comments:

Reviewer's Responses to Questions

**Comments to the Author**

1. Is the manuscript technically sound, and do the data support the conclusions?

Reviewer #1: Yes

Reviewer #2: Partly

2. Has the statistical analysis been performed appropriately and rigorously? 

Reviewer #1: Yes

Reviewer #2: Yes

3. Have the authors made all data underlying the findings in their manuscript fully available?

Reviewer #1: Yes

Reviewer #2: Yes

4. Is the manuscript presented in an intelligible fashion and written in standard English?

Reviewer #1: Yes

Reviewer #2: No

5. Review Comments to the Author

Reviewer #1: Thank you for the opportunity to review the manuscript entitled “Testosterone reduces uterine contractions in vivo: evidence for non-genomic action in rats.” The topic is potentially relevant to reproductive physiology, but several substantial issues in reporting, methodology, and interpretation need to be addressed before the manuscript can be considered for publication. My specific comments are as follows:

Abstract – Missing Sample Size and Group Details

The Abstract does not report any numerical information regarding the study population (number of animals included, size of the study and control groups, allocation, etc.). These details are essential for transparency and should be included to provide readers with a clear understanding of the sample structure.

Materials and Methods – Selection Criteria for Male Rats

The Methods section does not describe whether any criteria were used for selecting the male rats. It is important to specify age, weight, health status, breeding conditions, and any exclusion criteria, as these factors may influence hormonal and physiological responses.

Confounding Effect of General Anesthesia on Uterine Contractility

A major potential confounder is the assessment of uterine contractility during general anesthesia, as anesthetic agents may alter smooth muscle function and hormonal responsiveness. The authors should discuss the known impact of general anesthesia on uterine contractility and justify its use. Furthermore, it should be clarified why regional or local anesthesia was not considered, or whether such an approach would have been feasible in this experimental model.

Rationale for Intraperitoneal Administration of Testosterone

The rationale for administering testosterone intraperitoneally is not explained. The authors should clarify why this route was chosen over others (e.g., intravenous, subcutaneous), describe its pharmacokinetic implications, and discuss whether this method holds translational value for potential human studies. Without this clarification, the clinical relevance of the findings remains uncertain.

Results – Missing Numbers of Animals Included

The Results section does not report how many rats were actually included in the study or how many were allocated to each group. This information is fundamental for interpreting the statistical power and the reliability of the findings. The exact number of animals per group must be added.

Conclusion – Unsupported Clinical Recommendation

The concluding statement suggesting that testosterone should be used for the treatment of threatened preterm labor is inappropriate and not supported by the presented data. The study was conducted in rats, under experimental conditions, with unclear translational applicability. Such a statement should be removed, and the conclusion should be limited to the observed experimental findings without extrapolating to clinical practice.

Reviewer #2: Conceptual and Biological Framework:

The central hypothesis that testosterone acutely reduces uterine contractility in vivo is biologically plausible, but the mechanistic rationale is insufficiently developed, particularly regarding receptor-mediated vs non-genomic effects.

The manuscript does not clearly distinguish whether observed effects are direct uterine actions or secondary to systemic endocrine or autonomic modulation.

Experimental Design:

Estrous cycle control is inadequately addressed. Given the profound influence of cycle stage on uterine contractility and steroid responsiveness, failure to rigorously synchronize or stratify animals represents a major biological confounder.

The choice of testosterone dose and route of administration lacks sufficient justification in terms of physiological relevance. It is unclear whether serum levels achieved correspond to: physiological female exposure, pathological hyperandrogenism, or pharmacologic supraphysiological conditions.

The timing of outcome assessment relative to hormone administration is not clearly justified, limiting interpretation of genomic vs non-genomic effects.

Outcome Measures:

Uterine contractility is treated as a unitary endpoint, but: Frequency, amplitude, and coordination of contractions are not consistently analyzed or discussed as distinct physiological variables.

No direct assessment of myometrial androgen receptor expression or signaling is provided, weakening causal inference.

Controls and Comparators:

The absence (or limited use) of: estrogen-only controls, androgen receptor antagonists, or ovariectomized replacement models significantly limits mechanistic interpretation.

Vehicle controls are insufficiently discussed, particularly regarding stress and injection effects on uterine activity.

Interpretation of Results:

The discussion overstates causality, implying direct androgenic suppression of uterine contractility without adequately excluding indirect pathways.

Translational implications for human reproductive physiology are speculative and insufficiently qualified, given species-specific differences in myometrial steroid responsiveness.

Statistical Considerations:

Sample size justification is weak or absent.

Repeated-measures structure (if present) is not clearly accounted for, raising concerns about pseudoreplication.

Variability within treatment groups suggests biological heterogeneity that is not explored.

6. PLOS authors have the option to publish the peer review history of their article (what does this mean?). If published, this will include your full peer review and any attached files.). If published, this will include your full peer review and any attached files.

.

Reviewer #1: **Yes:**Basilio PecorinoBasilio Pecorino

Reviewer #2: **Yes:**JL Paz-IbarraJL Paz-Ibarra

---

## [Author Response · Author response to Decision Letter 1]

12 Feb 2026

The authors thank the Editor and the Reviewers for their useful questions and remarks which have contributed considerably to improving the quality of our manuscript. Our answers are given below.

Responses to the Editor

To comply with PLOS One submissions requirements, in your Methods section, please provide additional information regarding the experiments involving animals and ensure you have included details on (1) methods of sacrifice, (2) methods of anesthesia and/or analgesia, and (3) efforts to alleviate suffering.

For in vivo experiments, animals were anesthetized with a combination of ketamine and xylazine (36+4 mg/kg), as indicated in the first version of the manuscript, and then euthanized with a ten-fold dose (ketamine and xylazine 360+40 mg/kg). For in vitro experiments, the animals were euthanized in a CO2 chamber with gradually increasing carbon dioxide concentrations. The ways of euthanasia of the animals are also indicated in the revised version of the manuscript.

Thank you for stating the following financial disclosure: “Project No. TKP2021-EGA-32 was implemented with the support provided by the Ministry of Innovation and Technology of Hungary from the National Research, Development, and Innovation Fund, financed under the TKP2021-EGA funding scheme. The work was supported by the Stipendium Hungaricum Scholarship, University of Szeged, Hungary and by the Research Fund of Albert Szent-Györgyi Medical School, University of Szeged, Hungary. The work was also funded by the Research Fund of Albert Szent-Györgyi Medical School, University of Szeged, Hungary and University of Szeged Open Access Fund (Grant number 8261)” Please state what role the funders took in the study. If the funders had no role, please state: "The funders had no role in study design, data collection and analysis, decision to publish, or preparation of the manuscript." If this statement is not correct you must amend it as needed. Please include this amended Role of Funder statement in your cover letter; we will change the online submission form on your behalf.

We made the following statement: “The funders had no role in study design, data collection, analysis, decision to publish, or preparation of the manuscript.” This statement is included indicated in the revised manuscript.

We note that your Data Availability Statement is currently as follows: All relevant data are within the manuscript and its Supporting Information files. Please confirm at this time whether or not your submission contains all raw data required to replicate the results of your study. Authors must share the “minimal data set” for their submission.

All raw data necessary to replicate the results have been included as supplementary information in separated excel files.

Editor Comments

1. I am so sorry, but I could not understand the meaning of Dose 1 and Dose 2 in Figure 2.

Thank you for your comment. In this study, we examined the possible independent uterine effect of the administered solvent. The doses refer to 100 microliters of the administered solvent, which was administered twice when the antagonist was used. For ease of understanding, we have modified the notation to First and Second administration in the revised version of the manuscript.

2. Please provide in-brief description of experiment in the legend of figures and title of Tables as well as sample size and Data (Mean +/- SD).

The requested descriptions have been provided in the revised version of the manuscript.

3. Please show statistical differences for T levels between pregnant and non-pregnant rats in Figure 4; It could be useful to indicate the time of T injection by an arrow.

The statistical differences were presented, and the timing of testosterone injections was also added to the figure (Figure 5 in the revised manuscript) and discussed in the revised manuscript.

4. I would suggest to include the baseline HPPD level for comparisons shown in Figure 7

The NP and P controls represent baselines (Figure 9 of the revised manuscript), as presented in the first version of the manuscript.

5. Sensitivity and specificity of T and HPPD ELISA kits are missing; hence the detection ranges were provided.

The sensitivity of the ELISA kits is 18.75 pg/ml for both T and HPPD. Both are specific for testosterone or HPPD only, and there is no apparent cross-reactivities. This information has been included in the Methods section of the revised manuscript.

6. Please provide the method to avoid the blood coagulation to obtain the plasma.

A 1 ml blood sample was taken from the tail vein before and after dosing (0, 5, 15, 30, 60, 120, 240, and 480 minutes) into tubes containing K2EDTA (1 mg/tube) (BD Microtainer, Thermo Fisher Scientific Inc., Budapest, Hungary), then centrifuged (1700× g, 10 minutes, 4 °C) to isolate the plasma. This methodological description was also included in the revised version of the manuscript.

7. Please provide a single paragraph abstract

The single paragraph abstract was provided in the revised version of manuscript.

8. Please add significance of the study at the beginning of abstract

We modified the significance of the study based on the reviewers' comments and indicated this in the summary of the revised version of the manuscript.

Responses to Reviewer #1

1. The Abstract does not report any numerical information regarding the study population (number of animals included, size of the study and control groups, allocation, etc.). These details are essential for transparency and should be included to provide readers with a clear understanding of the sample structure.

The study population data were added to the abstract in the revised version of the manuscript. Female rats received only testosterone (3/10/30/100/300 mg/kg ip), flutamide (100 mg/kg ip), solvent (DMSO+Macrogol 25%+75%, 1 ml/kg ip) or normal saline (1 ml/kg) (n=5-8/group, 160-220 g). To measure testosterone and HPPD plasma levels, female rats weighed 200-220 g (n=4-8/group).

2. The Methods section does not describe whether any criteria were used for selecting the male rats. It is important to specify age, weight, health status, breeding conditions, and any exclusion criteria, as these factors may influence hormonal and physiological responses.

The study was conducted in female rats (160-220 g) to determine the effect of testosterone on the contractile capacity of non-pregnant and pregnant uterus. Pregnant animals were used on the last day of pregnancy (day 22 of pregnancy), while non-pregnant animals were selected based on the estrus phase of the estrous cycle to minimize differences caused by hormonal effects. Male rats were used for mating only and weighed between 240 and 260 g. All animals were healthy, with no diseases or adverse symptoms observed. Body weight and the term "healthy" were incorporated into the revised manuscript.

3. Confounding Effect of General Anesthesia on Uterine Contractility

A. A major potential confounder is the assessment of uterine contractility during general anesthesia, as anesthetic agents may alter smooth muscle function and hormonal responsiveness. The authors should discuss the known impact of general anesthesia on uterine contractility and justify its use.

It is known that general anesthetics relax the smooth muscles of the uterus, so we chose a combination of ketamine and xylazine, which causes moderate relaxation compared to other general anesthetics. Since we conducted a self-controlled experiment, any initial relaxation caused by the general anesthetic was part of the control phase in all cases.

B. Furthermore, it should be clarified why regional or local anesthesia was not considered, or whether such an approach would have been feasible in this experimental model.

General anesthesia is required to keep the animal’s unconscious to avoid muscle movements and changes in body position that would significantly interfere with the measurement of uterine contractions. Epidural or spinal or regional local anesthesia cannot be used because awake rats are unable to cooperate during measurement.

4. Rationale for Intraperitoneal Administration of Testosterone

A. The rationale for administering testosterone intraperitoneally is not explained. The authors should clarify why this route was chosen over others (e.g., intravenous, subcutaneous), describe its pharmacokinetic implications.

Testosterone is lipid-soluble (insoluble in water), so we dissolved it in a combination of DMSO and macrogol. Due to the toxicity of such solutions (e.g., hemolysis, risk of precipitation), they are not suitable for intravenous administration, so we chose the second fastest-acting method of administration, which was intraperitoneal administration.

B. Discuss whether this method holds translational value for potential human studies. Without this clarification, the clinical relevance of the findings remains uncertain.

Intraperitoneal administration of testosterone has no direct translational significance, as this type of administration is not used in clinical practice. However, our study provides the first evidence of the uterine-relaxing effect of testosterone in vivo during pregnancy. These results may therefore justify the development of new, more water-soluble testosterone analogues that allow for dosing methods that are already applicable in clinical practice.

5. Results – Missing Numbers of Animals Included

A. The Results section does not report how many rats were actually included in the study or how many were allocated to each group. This information is fundamental for interpreting the statistical power and the reliability of the findings. The exact number of animals per group must be added.

The number of rats per group is indicated in the revised version of the manuscript.

6. The concluding statement suggesting that testosterone should be used for the treatment of threatened preterm labor is inappropriate and not supported by the presented data. The study was conducted in rats, under experimental conditions, with unclear translational applicability. Such a statement should be removed, and the conclusion should be limited to the observed experimental findings without extrapolating to clinical practice.

Thank you for your comment. We agree with the reviewer that our translation conclusion in the first version of the manuscript was rather speculative, so we have modified our presumed translation significance to the following: " Testosterone or its analogues may be good candidates for further preclinical and clinical studies for the potential treatment of conditions associated with uterine hyperactivity."

Responses to Reviewer #2:

1. The central hypothesis that testosterone acutely reduces uterine contractility in vivo is biologically plausible, but the mechanistic rationale is insufficiently developed, particularly regarding receptor-mediated vs non-genomic effects.

We previously published an in vitro study in which testosterone induced a rapid non-genomic relaxing effect in the uterine smooth muscle of 22-day pregnant rats. The non-genomic effect was demonstrated by the ineffectiveness of the RNA transcription inhibitor actinomycin D and the protein synthesis inhibitor cycloheximide, which did not modify the effect of testosterone. [doi: 10.1016/j.lfs.2020.118584].

Furthermore, in our present study, we demonstrated that the 30-minute effect of testosterone did not induce any genomic response in vivo, as inferred from unchanged plasma 4-HPPD levels. The uterine relaxant effect of testosterone was not altered in the presence of an androgen receptor blocker (flutamide). Furthermore, during manuscript revision, we performed in vitro measurements to confirm the most likely non-genomic mechanism of action of testosterone via calcium channels. Testosterone (10-4 M) blocked calcium chloride-induced contractions in a calcium-free environment in a manner like nifedipine (10-7 M), strongly suggesting a role for calcium blockade in the non-genomic effects of testosterone. These new in vitro results have been incorporated into the revised version of the manuscript.

2. The manuscript does not clearly distinguish whether observed effects are direct uterine actions or secondary to systemic endocrine or autonomic modulation.

The effect of testosterone is most likely not related to secondary endocrine effects, as such endocrine effects take longer than the 30-minute time interval used in our studies. Furthermore, testosterone exerts a similar direct effect on the uterus in vitro, as we observed in our previous study [https://doi.org/10.1016/j.lfs.2020.118584].

We agree with the reviewer that testosterone may have autonomous effects on uterine contractions via calcium channels and G protein-coupled membrane receptors, but these targets are in smooth muscle cells, so it seems very difficult to clearly distinguish between the autonomic and uterine smooth muscle effects of testosterone, especially in vivo. Nevertheless, the reviewer's idea is excellent, and we will take it into account in our future studies.

3. Experimental Design:

A. Estrous cycle control is inadequately addressed. Given the profound influence of cycle stage on uterine contractility and steroid responsiveness, failure to rigorously synchronize or stratify animals represents a major biological confounder.

As described in the first version, all experiments on non-pregnant rats were performed during the estrus phase, and the rats were selected based on vaginal impedance using the Estrus Cycle Monitor (IM-01, MSB-MET Ltd., Balatonfüred, Hungary). For non-pregnant experiments, rats with vaginal impedance between 4.5 and 7.5 kΩ on the day of the experiment (estrus phase interval) were selected. This type of selection of non-pregnant animals is a well-established and widely accepted method for avoiding endocrine disrupting factors.

B. The choice of testosterone dose and route of administration lacks sufficient justification in terms of physiological relevance. It is unclear whether serum levels achieved correspond to: physiological female exposure, pathological hyperandrogenism, or pharmacologic supraphysiological conditions.

The testosterone dosage range was selected based on the literature on intraperitoneal systemic testosterone administration in male rats [https://doi.org/10.1016/j.bbr.2014.01.013], as no data were available on testosterone administration in female rats, especially during pregnancy. Testosterone administration significantly increased plasma testosterone levels in both non-pregnant and pregnant rats, resulting in a 10-30-fold increase after the highest dose (300 mg/kg). These serum levels are achievable with exogenous testosterone administration. This single testosterone administration was not intended to model any physiological, pathophysiological, or supra-physiological condition, but only to measure the dose-dependent uterine relaxing effect of testosterone. Our animal model is not related to any endocrine disease.

C. The timing of outcome assessment relative to hormone administration is not clearly justified, limiting interpretation of genomic vs non-genomic effects.

As we wrote in the first version of the manuscript, there is solid evidence supporting the link between timing and non-genomic effects, so we chose the time for evaluating the results (less than 30 minutes) to be insufficient for a genomic response to develop, and the effects obtained could be attributed to a non-genomic effect [https://doi.org/10.1155/2020/8849641, https://doi.org/10.1016/j.steroids.2019.108509]. The ineffectiveness of the testosterone antagonist flutamide also confirms that we obtained non-genomic responses.

4. Outcome Measures:

A. Uterine contractility is treated as a unitary endpoint, but: Frequency, amplitude, and coordination of contractions are not consistently analyzed or discussed as distinct physiological variables.

We chose AUC-based evaluation to determine contraction changes. Since the AUC change includes both frequency and amplitude changes, it provides a complex but simple parameter for characterizing uterine contractions: if the frequency or amplitude decreases, it means that uterine activity has decreased, which is sensitively detected by the AUC analysis. The strain gauge method cannot be used to measure th

---

## [Editor Report · Decision Letter 1]

19 Feb 2026

PONE-D-25-63242R1Testosterone reduces uterine contractions in vivo: evidence for non-genomic action in ratsPLOS One

Dear Dr. Gaspar,

Thank you for submitting your manuscript to PLOS ONE. After careful consideration, we feel that it has merit but does not fully meet PLOS ONE’s publication criteria as it currently stands. Therefore, we invite you to submit a revised version of the manuscript that addresses the points raised during the review process.

If applicable, we recommend that you deposit your laboratory protocols in protocols.io to enhance the reproducibility of your results. Protocols.io assigns your protocol its own identifier (DOI) so that it can be cited independently in the future. For instructions see: https://journals.plos.org/plosone/s/submission-guidelines#loc-laboratory-protocols. Additionally, PLOS ONE offers an option for publishing peer-reviewed Lab Protocol articles, which describe protocols hosted on protocols.io. Read more information on sharing protocols at . Additionally, PLOS ONE offers an option for publishing peer-reviewed Lab Protocol articles, which describe protocols hosted on protocols.io. Read more information on sharing protocols at https://plos.org/protocols?utm_medium=editorial-email&utm_source=authorletters&utm_campaign=protocols..

We look forward to receiving your revised manuscript.

Kind regards,

Sayyed Mohammad Hadi Alavi

Academic Editor

PLOS One

**Journal Requirements:**

**Additional Editor Comments:**

Dear Prof. Robert Gaspar, PharmD, PhD, DSc

Thank you very much for your revision. I am very much grateful to you for providing data as supplemental information.

After reviewing your revision; I found your MS needs another revision for publication. In addition to my comments listed below; I would very much appreciate it if you kindly to consider following suggestions.

1- As AR was not been assessed, I would suggest to remove “evidence for non-genomic action” from the title. Use of Flu as an antagonist for AR is not enough.

2- I would suggest to add a schematic cartoon summarizing T inhibition of uterine contraction. It could be cited in Conclusion as Fig. 10.

3- It would be great if you edite the English of MS. There are many errors.

4- Please add data availability statement.

5- I would suggest to use T as an abbreviation for testosterone after expanding in its location.

Thank you very much for choosing PLOS ONE to publish your work

Very best regards

Hadi Alavi

AE, PLOS ONE

Comments

Beginning of abstract: Please indicate novelty and strength of the present study at the beginning of the abstract. Abstract should provide the readers with significance of the study.

L20: This study aimed to investigate the non-genomic effect of testosterone (T) on uterine muscle contractility in non-pregnant and 22-day pregnant rats, in vivo.

L21: Circulating T levels were measured withing 8 hours after a single T administration (10 mg/kg ip) by ELISA.

L36: Please re-write: Testosterone plasma levels showed dose-dependency as well.

L37: remained unchanged in both 22-day non-pregnant and pregnant rats after 30 min of T administration.

L38: A single dose T induces

Testosterone is the most important hormone of the androgen family in both males and

L55: … females. In males, T is mainly produced in the Leydig cells of the testes (8,10).

L57: If you provide values for rats, would be perfect and meaningful with your experimental model.

L97: Please indicate the minimum number of animals derived from calculation.

L116: T, flutamide, and nifedipine were purchased from Sigma-Aldrich (Budapest, Hungary), DMSO was purchased from Fisher Scientific (Loughborough, UK), and Macrogol 400 was purchased from MAGIlab Ltd. (Budapest, Hungary). The rat T and hydroxyphenylpyruvate dioxygenase (HPPD) ELISA kits were purchased from Wuhan Fine Biotech Co. Ltd. (Wuhan, China).

L121: 2.4. Plasma T assessment

L122: intraperitoneal administration (ip)

L125: It may be omitted: “it is specific for testosterone only and there are no apparent cross-reactivities.“

L129: for the plasma

L159: Similar to L122 (aforementioned comment”

L160-165: It was aforementioned for T assessment. I would suggest to merge T and HPPD assessments to avoid redundancy.

L193: ip administration of ketamine + xylazine (36 + 4 mg/kg in 20 ml solution) in a dose of 5 ml/kg.

L196: into 4 experimental groups, including (1) solvent control, (2) T, (3) T + flutamide, and (4) absolute control (n=5-8 per group).

L198: ip dose of

L199: (100 mg/kg). The solvent

L200: l/kg ip). The …. Physiological saline

L134, 153, 164, 206: If all values are mean +/- SD; please avoid reducnancy and indicate it in the “Statistical analysis section).

Figure 2: ip should be in small letters.

Figure 4: Please clarify whether T1 and T2 referrers to Time 1 and Time 2, respectively. I would suggest a revision for x-axis of panels C and D.

L276: Fig. 8

L282: Please expand Emax and ED50 in the title of Table 2

L292: Fig. 9

Very importantly; please uniform 4-HPPD or HPPD through the manuscript.

Please use “T” as a substitute for testosterone through the manuscript, after expanding the abbreviation in the first position.

L324: ip >>> Please use “ip” as a substitute for intraperitoneal through the manuscript, after expanding the abbreviation in the first position.

Please add data availability statement.

---

## [Author Response · Author response to Decision Letter 2]

26 Feb 2026

Editor Comments:

1- As AR was not been assessed, I would suggest to remove “evidence for non-genomic action” from the title. Use of Flu as an antagonist for AR is not enough.

Thank you for your comment, but we believe it is better to keep the title as it is. We demonstrated that 30 minutes of testosterone exposure did not elicit any genomic response in vivo, as inferred from unchanged plasma 4-HPPD levels, so we did not rely solely on the androgen antagonist-resistant effect. In addition, we previously published an in vitro study in which testosterone elicited a rapid, non-genomic relaxing effect in the uterine smooth muscle of 22-day-old pregnant rats. The non-genomic effect was demonstrated by the ineffectiveness of the RNA transcription inhibitor actinomycin D and the protein synthesis inhibitor cycloheximide, which did not modify the effect of testosterone. [doi: 10.1016/j.lfs.2020.118584].

2- I would suggest to add a schematic cartoon summarizing T inhibition of uterine contraction. It could be cited in Conclusion as Fig. 10.

Thank you for this thoughtful comment; it was considered, and Fig 10 was added in the revised version of the manuscript.

3- It would be great if you edit the English of MS. There are many errors.

Thank you for this advice, it was considered and the English of the manuscript was edited in the revised version.

4- Please add data availability statement.

We added it in the revised version of the manuscript.

5- I would suggest to use T as an abbreviation for testosterone after expanding in its location.

The abbreviations were considered throughout the manuscript in the revised version

6. All other comments were considered and edited in the revised version of the manuscript.

---

## [Editor Report · Decision Letter 2]

2 Mar 2026

PONE-D-25-63242R2Testosterone reduces uterine contractions in vivo: evidence for non-genomic action in ratsPLOS One

Dear Dr. Gaspar,

Thank you for submitting your manuscript to PLOS ONE. After careful consideration, we feel that it has merit but does not fully meet PLOS ONE’s publication criteria as it currently stands. Therefore, we invite you to submit a revised version of the manuscript that addresses the points raised during the review process.

If applicable, we recommend that you deposit your laboratory protocols in protocols.io to enhance the reproducibility of your results. Protocols.io assigns your protocol its own identifier (DOI) so that it can be cited independently in the future. For instructions see: https://journals.plos.org/plosone/s/submission-guidelines#loc-laboratory-protocols. Additionally, PLOS ONE offers an option for publishing peer-reviewed Lab Protocol articles, which describe protocols hosted on protocols.io. Read more information on sharing protocols at . Additionally, PLOS ONE offers an option for publishing peer-reviewed Lab Protocol articles, which describe protocols hosted on protocols.io. Read more information on sharing protocols at https://plos.org/protocols?utm_medium=editorial-email&utm_source=authorletters&utm_campaign=protocols..

We look forward to receiving your revised manuscript.

Kind regards,

Sayyed Mohammad Hadi Alavi

Academic Editor

PLOS One

Journal Requirements:

Additional Editor Comments :

Thank you very much for your revision. I have seen your revision, however I am so sorry to return the MS back to you for another revision.

The revision should follow the PLOS ONE instruction.

1- Supplementary materials: I do not know why S1 is titled Fig 3? or S2 is Fig4? Usually, S1 should be Fig 1, S2 should be Fig 2, S3 Table 1, S4 Table 2, S4 Fig 3 and so on.

2- It is not acceptable to respond "All other comments were considered and edited in the revised version of the manuscript". The letter of response should answer all comments point-by-point.

3- Please provide an in-brief description for the representative schismatic of Figure 10. If some pathways in the schematic have been previously investigated, please provide list of references that support the pathways shown in the schematic.

4- For showing track changes, Please use the Office Platform to compare the past version as a source (in your case R1) and the revised version (in your case R2). This will help to see all details perfoemed during revision.

Thank you very much for your helps to our review quality and kind understandings

Hadi Alavi

AE, PLOS ONE

---

## [Author Response · Author response to Decision Letter 3]

2 Mar 2026

Journal Requirements:

There was no such comment.

The reference list is complete; we do not cite any retracted article.

Additional Editor Comments:

1- Supplementary materials: I do not know why S1 is titled Fig 3? or S2 is Fig4? Usually, S1 should be Fig 1, S2 should be Fig 2, S3 Table 1, S4 Table 2, S4 Fig 3 and so on.

Thank you for the comment. The fact is that Fig 1 and Fig 2 do not have any data to put in supplementary files (representative or schematic figures). Thus, the first figure that has supplementary material is Fig 3.

2- It is not acceptable to respond "All other comments were considered and edited in the revised version of the manuscript". The letter of response should answer all comments point-by-point.

L20: This study aimed to investigate the non-genomic effect of testosterone (T) on uterine muscle contractility in non-pregnant and 22-day pregnant rats, in vivo.

Thank you, we accepted the recommendation and the sentence is involved in the revised version of the manuscript as “This study aimed to investigate the non-genomic effect of testosterone (T) on uterine muscle contractility in non-pregnant and 22nd day pregnant rats, in vivo”.

L21: Circulating T levels were measured withing 8 hours after a single T administration (10 mg/kg ip) by ELISA.

Thank you, we accepted the recommendation and the sentence is involved in the revised version of the manuscript as “Circulating T levels were measured within 8 hours after a single intraperitoneal (ip) administration of T (10 mg/kg) by ELISA”.

L36: Please re-write: Testosterone plasma levels showed dose-dependency as well.

Thank you, we accepted the recommendation and the sentence is involved in the revised version of the manuscript as “T plasma levels were proportional with the administered doses”.

L37: remained unchanged in both 22-day non-pregnant and pregnant rats after 30 min of T administration

Thank you, we accepted the recommendation and the sentence is involved in the revised version of the manuscript as “remained unchanged in both non-pregnant and 22nd day pregnant rats after 30 min of T administration”

L38: A single dose T induces

Thank you, we accepted the recommendation and the sentence is involved in the revised version of the manuscript as “A single dose T induces…”

Testosterone is the most important hormone of the androgen family in both males and

L55: … females. In males, T is mainly produced in the Leydig cells of the testes (8,10).

Testosterone (T) is the most important hormone in the androgen family in both males and Thank you, we accepted the recommendation and the sentence is involved in the revised version of the manuscript as “females, in males produced mainly in the Leydig cells of the testes”

L57: If you provide values for rats, would be perfect and meaningful with your experimental model.

Thank you, we accepted the recommendation and the sentence is involved in the revised version of the manuscript as “Plasma T levels in males are higher than in females (2,7,11) or in pregnant women (4,11)”

L97: Please indicate the minimum number of animals derived from calculation

Thank you, we accepted the recommendation and the sentence is involved in the revised version of the manuscript as follow “The minimum number of animals per group was 4”

L116: T, flutamide, and nifedipine were purchased from Sigma-Aldrich (Budapest, Hungary), DMSO was purchased from Fisher Scientific (Loughborough, UK), and Macrogol 400 was purchased from MAGIlab Ltd. (Budapest, Hungary). The rat T and hydroxyphenylpyruvate dioxygenase (HPPD) ELISA kits were purchased from Wuhan Fine Biotech Co. Ltd. (Wuhan, China).

Thank you, we accepted the recommendation and the sentence is involved in the revised version of the manuscript as “T, flutamide, and nifedipine were purchased from Sigma-Aldrich (Budapest, Hungary), DMSO was from Fisher Scientific (Loughborough, UK), Macrogol 400 was purchased from MAGIlab Ltd. (Budapest, Hungary), while Rat T and 4-HPPD ELISA kits were delivered by Wuhan Fine Biotech Co., Ltd. (Wuhan, China)”.

L121: 2.4. Plasma T assessment

Thank you, we accepted the recommendation, the 4-HPPD method was merged with testosterone, and the subtitle is involved in the revised version of the manuscript as “Plasma T and 4-HPPD assessment”

L122: intraperitoneal administration (ip)

Thank you, we accepted the recommendation and the sentence is involved in the revised version of the manuscript as “intraperitoneal (ip) administration”

L125: It may be omitted: “it is specific for testosterone only and there are no apparent cross-reactivities.“

Thank you, we accepted the recommendation and the sentence was omitted.

L129: for the plasma

Thank you, we accepted the recommendation and the sentence is involved in the revised version of the manuscript as “…for the plasma”

L159: Similar to L122 (aforementioned comment”

Thank you, we accepted the recommendation and the sentence is involved in the revised version of the manuscript as “..ip administration”

L160-165: It was aforementioned for T assessment. I would suggest to merge T and HPPD assessments to avoid redundancy.

Thank you, we accepted the recommendation and the sentences are involved in the revised version of the manuscript as “Additionally, 4-HPPD levels were measured before (baseline) and 30-minutes after the administration of the T doses (100 or 300 mg/kg) for both non-pregnant estrus rats and 22nd day pregnant (n=6-8/group) using a commercial rat 4-HPPD Enzyme Immunoassay Kit (Wuhan Fine Biotech Co., Ltd., China), with a detection range between 31.25-2000 pg/ml and sensitivity of 18.75 pg/ml”.

L193: ip administration of ketamine + xylazine (36 + 4 mg/kg in 20 ml solution) in a dose of 5 ml/kg.

Thank you, we accepted the recommendation and the sentence is involved in the revised version of the manuscript as “ip administration of ketamine + xylazine (36+4 mg/kg in 20 ml solution) in a dose of 5 ml/kg”.

L196: into 4 experimental groups, including (1) solvent control, (2) T, (3) T + flutamide, and (4) absolute control (n=5-8 per group).

Thank you, we accepted the recommendation and the sentence is involved in the revised version of the manuscript as “…..into 4 experimental groups including (1) solvent control, (2) T, (3) T + flutamide, (4) absolute control (n=5-8 per group)”.

L198: ip dose of

Thank you, we accepted the recommendation and the sentence is involved in the revised version of the manuscript as “..ip dose of”.

L199: (100 mg/kg). The solvent

Thank you, we accepted the recommendation and the sentence is involved in the revised version of the manuscript as “(100 mg/kg). The solvent..”

L200: l/kg ip). The …. Physiological saline

Thank you, we accepted the recommendation and the sentence is involved in the revised version of the manuscript as “The absolute control group received physiological saline”

L134, 153, 164, 206: If all values are mean +/- SD; please avoid reducnancy and indicate it in the “Statistical analysis section”.

Thank you, we accepted the recommendation they were deleted and added to the statistical analysis section, the sentence is involved in the revised version of the manuscript as “All data are expressed as means ± standard deviation (SD)”.

Figure 2: ip should be in small letters.

Thank you, we accepted the recommendation and the “ip” abbreviation are in small letter in the revised figure in the revised version of the manuscript.

Figure 4: Please clarify whether T1 and T2 referrers to Time 1 and Time 2, respectively. I would suggest a revision for x-axis of panels C and D.

Thank you, we accepted the recommendation and it is “Time 1” and “Time 2” in the revised version of the manuscript.

L276: Fig. 8

Thank you, we accepted the recommendation and it is involved in the revised version of the manuscript as “Fig. 8”

L282: Please expand Emax and ED50 in the title of Table 2

Thank you, we accept the recommendation and the table was corrected and is involved in the revised version of the manuscript as:

Parameter Non-pregnant rats 22nd day pregnant rats

T T+Flut T T+Flut

Emax (%) 67.5 ± 5.9 72.9 ± 7.2 44.4 ± 5.9** 42.8 ± 6.9**

ED50 (mg/kg) 11.7 ± 1.9 12.45 ± 1.8 14.3 ± 2.3 11.5 ± 2.5

L292: Fig. 9

Thank you, we accepted the recommendation and it is involved in the revised version of the manuscript as “Fig. 9”

Very importantly; please uniform 4-HPPD or HPPD through the manuscript.

Thank you, we accepted the recommendation and it is uniformed in the revised version of manuscript as “4-HPPD”.

Please use “T” as a substitute for testosterone through the manuscript, after expanding the abbreviation in the first position.

Thank you, we accepted the recommendation and it is substituted to “T” through the manuscripts in the revised version.

L324: ip >>> Please use “ip” as a substitute for intraperitoneal through the manuscript, after expanding the abbreviation in the first position.

Thank you, we accepted the recommendation and it is “ip” throughout the manuscript in the revised version

Please add a data availability statement.

Thank you, we accepted the recommendation and the data are included in the revised version of the manuscript as follow:

Supporting information

S1 Fig 3- The in vivo effect of solvent on the contraction of rat uterine muscle.

S2 Fig 4- The effect of time passage on the contraction of the rat uterine muscle in vivo.

S3 Fig 5- Plasma T concentration-time curves after a single ip injection (10 mg/kg) in non-pregnant and 22nd day pregnant rats.

S4 Fig 6- Plasma T levels before and 30 min after the single dose of T administration in non-pregnant and 22nd day pregnant rats.

S5 Fig 7- The inhibitory effect of T and nifedipine on non-pregnant and 22nd day pregnant uterine tissues in vitro.

S6 Fig 8- The non-genomic uterine relaxing effect of T in vivo.

S7 Fig 9- Plasma levels of 4-HPPD after T administration.

3- Please provide an in-brief description for the representative schismatic of Figure 10. If some pathways in the schematic have been previously investigated, please provide a list of references that support the pathways shown in the schematic.

Thank you, we accepted the recommendation and the sentences are involved in the revised version of the manuscript as:

“Schematic diagram summarizing the effect of T on inhibiting uterine contraction. The red line represents the inhibition demonstrated in our experiment, while the blue (inhibition) and green (activation) lines represent the mechanisms described in previous studies. T inhibits voltage dependent Ca2+-channels having a crucial role in its uterine relaxing effect (red line). Furtherly, T stimulates the 7TM (G-protein coupled) receptors (short green arrow) and increases cAMP level of the uterine tissues which inhibits the Ca2+channels activity (blue line) (32). Additionally, T activates different K+ channel types in smooth muscle (long green arrow), that also leads to relaxation by reducing intracellular K+ level (33). AC: Adenylyl cyclase, ATP: adenosine triphosphate, cAMP: cyclic adenosine monophosphate, T: testosterone, 7TM: seven-transmembrane receptor (G-protein coupled receptor)”.

4- For showing track changes, please use the Office Platform to compare the past version as a source (in your case R1) and the revised version (in your case, R2). This will help to see all details perfoemed during revision.

The revised version is trackable now as it was requested.

---

## [Editor Report · Decision Letter 3]

15 Apr 2026

Testosterone reduces uterine contractions in vivo: evidence for non-genomic action in rats

PONE-D-25-63242R3

Dear Dr. Gaspar,

We’re pleased to inform you that your manuscript has been judged scientifically suitable for publication and will be formally accepted for publication once it meets all outstanding technical requirements.

An invoice will be generated when your article is formally accepted. Please note, if your institution has a publishing partnership with PLOS and your article meets the relevant criteria, all or part of your publication costs will be covered. Please make sure your user information is up-to-date by logging into Editorial Manager at Editorial Manager® and clicking the ‘Update My Information' link at the top of the page. For questions related to billing, please contact  and clicking the ‘Update My Information' link at the top of the page. For questions related to billing, please contact billing support..

Kind regards,

James J Cray Jr., Ph.D.

Academic Editor

PLOS One
---

## [Editor Report · Acceptance letter]

PONE-D-25-63242R3

PLOS One

Dear Dr. Gaspar,

I'm pleased to inform you that your manuscript has been deemed suitable for publication in PLOS One. Congratulations! Your manuscript is now being handed over to our production team.

Kind regards,

on behalf of

Dr. James J Cray Jr.

Academic Editor

PLOS One